# Predictive Role of Complete Blood Count-Derived Inflammation Indices and Optical Coherence Tomography Biomarkers for Early Response to Intravitreal Anti-VEGF in Diabetic Macular Edema

**DOI:** 10.3390/biomedicines13061308

**Published:** 2025-05-27

**Authors:** Ece Ergin, Ana Maria Dascalu, Daniela Stana, Laura Carina Tribus, Andreea Letitia Arsene, Marina Ionela Nedea, Dragos Serban, Claudiu Eduard Nistor, Corneliu Tudor, Dan Dumitrescu, Paul Lorin Stoica, Bogdan Mihai Cristea

**Affiliations:** 1Doctoral School, “Carol Davila” University of Medicine and Pharmacy Bucharest, 020021 Bucharest, Romania; ece.ergin@drd.umfcd.ro (E.E.); dragos.serban@umfcd.ro (D.S.);; 2Ophthalmology Department, Emergency University Hospital Bucharest, 050098 Bucharest, Romania; 3Faculty of Medicine, “Carol Davila” University of Medicine and Pharmacy Bucharest, 020021 Bucharest, Romania; 4Faculty of Dental Medicine, “Carol Davila” University of Medicine and Pharmacy Bucharest, 020021 Bucharest, Romania; 5Faculty of Pharmacy, “Carol Davila” University of Medicine and Pharmacy Bucharest, 020021 Bucharest, Romania; 6“Mina Minovici” Institute of Legal Medicine, 077160 Bucharest, Romania

**Keywords:** diabetic macular edema, intravitreal anti-VEGF, neutrophil-to-lymphocyte ratio, systemic inflammation index, platelet-to-lymphocyte ratio, OCT biomarkers, intraretinal cysts, hyperreflective retinal spots, disorganization of retinal inner layers, central macular thickness

## Abstract

**Background:** Diabetic macular edema (DME) is the leading cause of vision impairment in diabetic patients, with intravitreal anti-vascular endothelial growth factor (anti-VEGF) injections being the first-line therapy. However, one-third of patients exhibit persistent DME despite treatment, suggesting additional pathogenic factors. This study aimed to evaluate the predictive value of complete blood count (CBC)-based inflammation indexes and optical coherence tomography (OCT) parameters in determining early anti-VEGF treatment effectiveness in DME. **Methods:** One hundred and four naïve patients with DME, treated with 0.05 mL of intravitreal aflibercept were retrospectively analyzed. Blood parameters analyzed included neutrophil-to-lymphocyte ratio (NLR), monocyte-to-lymphocyte ratio (MLR), platelet-to-lymphocyte ratio (PLR), and systemic immune-inflammation index (SII). Baseline OCT biomarkers included subretinal fluid (SRF), intraretinal cysts (IRC), hyperreflective retinal spots (HRS), and disorganization of retinal inner layers (DRIL). Treatment response was defined as a minimum 10% reduction in central macular thickness (CMT) at one month post-injection. **Results:** NLR, MLR, PLR, and SII were significantly higher in non-responders (*p* < 0.001), but their predictive value was fair, with an area under the ROC curve ranging between 0.704 (MLR) and 0.788 (SII). A multivariate model including SII, initial CMT, and the presence of IRC showed an excellent prediction value for early anatomical response (AUC ROC of 0.911). At the same time, lower PLR, DRIL, SRF, and the absence of HRF were correlated with early gain in BCVA. **Conclusions:** CBC-derived inflammation indices and OCT biomarkers have prognostic value in predicting early response to anti-VEGF therapy in DME in terms of functional and anatomical outcomes. These findings could help identify poor responders and guide personalized treatment strategies.

## 1. Introduction

Diabetic macular edema (DME) is among the leading causes of vision impairment in individuals with diabetes, profoundly affecting their quality of life and imposing significant demands on global healthcare systems [1,2]. It is characterized by fluid accumulation in the macula due to retinal microvascular damage, which is influenced by systemic metabolic dysfunction, vascular abnormalities, and chronic inflammation [3]. The increasing global prevalence of diabetes further exacerbates the burden of DME, with recent estimates showing 537 million adults with diabetes in 2021—a figure projected to rise to 783 million by 2045 [3,4].

Among available treatments, intravitreal anti-vascular endothelial growth factor (VEGF) agents, such as aflibercept, are well established as first-line therapies for managing DME. These agents target VEGF, a key mediator of pathological angiogenesis and vascular leakage, thereby mitigating macular edema and preserving visual acuity [5,6]. However, response to anti-VEGF therapy varies, with several studies reporting a favorable response to anti-VEGF therapy ranging between 31.5 and 62% [5,7,8,9]. Mehta et al. [7] analyzed real-world outcomes from over 40,000 eyes treated for DME, highlighting variability in anatomical and functional improvements across patients. Similarly, Nguyen et al. [10] emphasized variability in long-term outcomes based on individual baseline characteristics and disease severity. This variability underscores the need for predictive factors that may guide personalized treatment regimens. A recent study of Camacho and col. [11] revealed that the variation in response to anti-VEGF therapy encountered in patients with DME, despite similar metabolic control and disease duration, might be linked with epigenetics alterations related to metabolic memory. The authors found consistent evidence of an upregulation in DNMT1 expression in non-responders, which could be related to lipidic profile and pro-inflammatory signals [11].

The recommended dose for intravitreal aflibercept is 2 mg administrated every 4 weeks for the first 5 months, followed by bi-monthly injections. According to DRCR.net Protocol I, the response to treatment includes CMT reduction and/or best-corrected visual acuity (BCVA) improvements at least 180 days after treatment initiation. However, while consistent visual gain may be experienced up to follow-up at 2 years, even in patients with sub-optimal initial response [12], multiple studies found that early improvements observed at 3-month follow-up are significantly associated with medium- and long-term outcomes [13,14]. Moreover, several researchers found that changes in CMT and BCVA after one month from the first anti-VEGF injection correlate well with the 3-month response in individuals [15,16], predicting further expectations, as well as the need for therapeutic alternatives, such as intravitreal corticosteroids or laser therapy.

Previous research has investigated the predictive role of ocular and systemic parameters, including baseline central macular thickness (CMT), glycated hemoglobin (HbA1c) levels, and diabetes duration, in determining treatment outcomes. Several studies found that higher baseline CMT is linked to more favorable anatomical responses, while poor glycemic control and prolonged diabetes duration are associated with suboptimal results [11,12]. Inflammatory processes related to glucose and lipid metabolism disorders play a significant role in retinal microvascular damage, with diabetic retinopathy (DR) increasingly viewed as a chronic inflammatory condition [17,18]. Systemic biomarkers such as the neutrophil-to-lymphocyte ratio (NLR), monocyte-to-lymphocyte ratio (MLR), and platelet-to-lymphocyte ratio (PLR) have demonstrated utility in various conditions, including malignancies, cardiovascular diseases, and diabetes complications [19,20,21,22].

Additionally, retinal biomarkers such as hyperreflective foci and disorganization of the retinal inner layers (DRIL) were studied for their possible predictive role in therapeutic response [23,24]. However, the interplay between these factors and their relative contributions to early treatment success remains incompletely understood.

This study aims to analyze the correlations between complete blood count-based inflammation indices and optical coherence tomography (OCT) factors that correlate with early therapeutic response to intravitreal aflibercept in patients with DME, at one-month follow-up. By investigating these relationships and their impact on early clinical improvement, we seek to identify actionable predictive factors that can enhance clinical decision-making, support personalized treatment strategies, and improve outcomes for patients with DME.

## 2. Materials and Methods

### 2.1. Study Design and Setting

This retrospective analysis was carried out on treatment-naïve DME patients treated by intravitreal aflibercept in the Ophthalmology Department of the Emergency University Hospital of Bucharest, Romania, between January 2022 and January 2025. This study was approved by the Medical Ethics Committee of the Emergency University Hospital of Bucharest (Nr. 3029/16.01.2025) and was conducted in accordance with the ethical principles of the Declaration of Helsinki.

### 2.2. Inclusion Criteria

The study included patients aged 18 years or older with Type 2 diabetes mellitus. Participants were required to present treatment-naïve, clinically significant central-involved DME associated with vision loss, diagnosed clinically and confirmed by the presence of fovea thickening of ≥250 µm by OCT macular scan and/or the presence of specific changes in the macular architecture.

### 2.3. Evaluation Protocol

All patients underwent a comprehensive ophthalmological examination, which included evaluation of the best-corrected visual acuity (BCVA), intraocular pressure (IOP), slit lamp exam of the anterior segment of the eye, and a fundus evaluation. BCVA was assessed using the Snellen chart. Grading of the DR was performed according to the International Classification of Diabetic Retinopathy (ICDR) severity scale [25] by 2 independent examinators. Any disagreement was resolved by discussion.

OCT was conducted using the CIRRUS^®^ 5000 OCT system (Carl Zeiss^®^, Oberkochen, Germany) Macula Cube 512 × 128 program. The European School for Advanced Studies in Ophthalmology(ESASO) Classification of DME was used to characterize the types of DME encountered in the study group [26,27,28]. Descriptive parameters related to the integrity of the ellipsoidal zone (EZ) and external limiting membrane (ELM), the presence of subretinal fluid (SRF), intraretinal cysts (IRCs), DRIL, hyperreflective retinal spots (HRSs), vitreo-retinal interaction, and numeric data for CMT were documented before the first intravitreal anti-VEGF injection and at one-month follow-up.

Peripheral blood sampling was performed in the fasting state before the first intravitreal anti-VEGF injection, including a complete blood count (CBC) with differentials, blood glucose, urea, and creatinine. Systemic inflammatory indices were calculated such as the ratio of the total neutrophil count/total lymphocyte count (NLR), total platelet count/total lymphocyte count (PLR), and monocytes/platelets count (MLR), respectively, measured from the same blood test, using automated hematology analyzing devices (DxH800 Hematology Analyzer, Beckman Coulter, Istanbul, Turkey), and expressed in × 10^3^ cells/μL. SII (systemic immune-inflammation index) was calculated using the following formula: SII = platelet count × neutrophil count/lymphocyte count.

All selected patients were diagnosed and treated according to current guidelines of DME management [29]. For all patients admitted during the COVID-19 pandemic, strict measures for social distancing, mask-wearing, and decontamination of the surfaces were observed to prevent in-hospital infections with SARS-CoV-2 [30,31].

A dose of 0.05 mL of aflibercept was administered by intravitreal injection, using a 28 G fine needle, in the operating room, on a daily admission. Written informed consent was obtained before the procedure after reasonable disclosure [32,33]. Patients were examined 4 weeks (±1) after the first intravitreal injection where BCVA, IOP, dilated fundus ophthalmoscopy, and OCT were performed. All relevant demographic and ocular parameters mentioned above were retrospectively collected from medical records.

### 2.4. Exclusion Criteria

Patients were excluded from the study if they had systemic conditions such as hematologic, oncologic, inflammatory, or infectious diseases, or a history of thromboembolic events, including acute coronary syndrome or stroke, within the past 3 months. Ocular exclusion criteria included vitreous hemorrhage before intravitreal treatment, vitreous opacities, or cataracts significantly impairing the visualization of the ocular fundus and the quality of OCT images. Additionally, patients with retinal conditions that could interfere with the assessment of DR progression, such as retinal detachment, age-related macular degeneration, retinal vascular obstructions, inherited retinal diseases, chorioretinitis (active or scarring), uveitis, or central areolar choroid dystrophy, were excluded. Other exclusion criteria included a history of eye surgery within the previous 6 months, prior vitreoretinal surgery, silicone oil in the eye, or laser photocoagulation therapy. Patients with glaucoma, acute conjunctivitis, or blepharitis were also excluded. Finally, individuals using systemic or topical non-steroidal anti-inflammatory drugs or steroids were deemed ineligible for participation.

### 2.5. Outcome Measures

The primary outcome measure was a change in central macular thickness (CMT) measured by OCT one month after the first intravitreal anti-VEGF injection. Patients were categorized as early ‘responders’ if a reduction in CMT of 10% or greater from baseline was achieved at one month, and as ‘non-responders’ if there was less than a 10% reduction or an increase in CMT. The secondary outcome was the mean change in BCVA at one month to determine early “functional response”. For individuals undergoing bilateral anti-VEGF injections, the eye demonstrating a better response to treatment was selected for inclusion in the study. Ocular adverse events were carefully documented and reported [33,34,35,36] during the postoperative follow-up visits on day 1, day 7, and day 30.

### 2.6. Statistical Analysis

Numeric variables were expressed as mean (±SD), while categorical variables were presented as absolute and relative frequencies (%). Patients were categorized into two groups based on responder status. Group comparability was assessed by comparing baseline demographic data and follow-up duration between groups. The normality and heteroskedasticity of continuous data were evaluated using the Shapiro–Wilk test and Levene’s test, respectively. Continuous variables were compared using the unpaired Student’s *t*-test, Welch’s *t*-test (for unequal variances), or the Mann–Whitney U test, depending on data distribution. Categorical outcomes were compared using the chi-squared test or Fisher’s exact test, as appropriate. A two-tailed alpha level of 0.05 was considered statistically significant. A *p*-value < 0.05 was considered statistically significant.

Multivariate regression analysis including all relevant variables was performed to identify the best predictive model for early anatomical response. A different regression analysis was carried out for the factors that correlated best with the improvements in BCVA (early functional responders). Statistical analysis was performed with EasyMedStat (version 3.38; www.easymedstat.com, accessed 15 February 2025) and MedCalc Software (v23.1.7, © 2025 MedCalcSoftwareLtd., Ostend, Belgium).

## 3. Results

A total number of 104 patients, aged between 47 and 78 years, were included in the study. Out of these, 66 (63.46%) presented a decrease in CMT of more than 10% at 4 weeks of evaluation (early responders group), while the other 38 (36.53%) were considered non-responders. The demographic, clinical, and biochemical characteristics of the study groups are presented in Table 1.

No significant differences were observed between the groups in terms of gender or age. The total neutrophil count was significantly lower (*p* = 0.005) and the lymphocyte count was significantly higher (*p* = 0.005) in the responder group compared to the non-responder group. Moreover, all systemic inflammatory indices were significantly lower in the early responder group: NLR (2.29 ± 0.84 vs. 3.4 ± 1.44; *p* < 0.001), MLR (0.28 ± 0.09 vs. 0.39 ± 0.16; *p* < 0.001), PLR (121.39 ± 38.35 vs. 164.92 ± 71.46; *p* < 0.001), and SII (546.86 ± 183.27 vs. 886.44 ± 407.06; *p* < 0.001). Additionally, pairwise analysis revealed no significant differences in serum glucose, urea, or creatinine levels between the two groups.

### 3.1. Comparative Ophthalmologic and OCT Findings in Initial and Follow-Up Exam

The patients included in the early responder group had a higher baseline CMT (*p* < 0.001) and a lower baseline BCVA (*p* = 0.012) compared to non-responders (Table 2).

The mean BCVA change was higher in the early responders group (0.15 ± 0.17 vs. 0.06 ± 0.16). However, a significant decrease in CMT was associated with vision improvement in only 63.6%. On the other hand, an improvement in BCVA was encountered at 26.3% in the non-responder group, suggesting that structure and function are not always superposed in managing DME.

Most of the patients included in the study presented advanced DME according to ESASO Classification (40 patients, 57.6%), while 19.2% (20 patients) were diagnosed with early DME and 23.1% (24 patients) severe DME, with disruption of ELM and inner retinal layer disorganization. The Chi-squared test and Chi-squared test for trend showed a significant correlation for trend between ESASO grade and post-injection decrease in CMT (*p* <0.001, Table 2).

However, there were fewer patients with severe DME that experienced improvements in BCVA, despite significant decreases in CMT. One explanation may be due to chronic long-term changes that led to anatomical disruption of the adherent junctions between the Müller cells and photoreceptors, macular ischemia, and worse visual outcomes [26]; see Figure 1.

The post-injection mean CMTs were lower in the responder group; however, the differences were not significant (*p* = 0.367). In the responders group, CMT decrease varied widely with a mean value of −173.03 µm, being well correlated with the initial CMT value (*p* < 0.001). Furthermore, SRF and IRC were more prevalent in the responders than in the non-responders (*p* = 0.001; *p* = 0.004, respectively). DRIL was more frequent in responders (81.82%) than in non-responders (63.16%), but the differences were not statistically significant (*p* = 0.06).

No significant adverse events were reported during the follow-up period. Figure 2 and Figure 3 illustrate different responses in patients with advanced DME.

### 3.2. Predictors of Early Anti-VEGF Treatment Response in Study Group

Lower initial values of the systemic inflammatory biomarkers (NLR, PLR, MLR, and SII) were well correlated with early therapeutic response in patients with DME (*p* <0.001). However, their predictive value was fair, with a better AUC ROC being observed for SII (0.788) and NLR (0.778) (Figure 4, Table 3).

When evaluating OCT biomarkers, initial CMT was strongly associated with early anatomic response (*p* <0.001). Patients with subretinal fluid (SRF) showed a significantly greater reduction in CMT, with a median CMT decrease of 32.28% in patients with SRF and only 12.76 in patients without SRF (*p* < 0.001). Similarly, the presence of intraretinal cysts (IRCs) was associated with a significantly greater reduction in CMT after anti-VEGF therapy, with a median decrease of 38.51% vs. 12.5% in patients not presenting this OCT biomarker at the initial evaluation (*p* < 0.001). However, only initial CMT presents a good predictive value (AUC ROC 0.818) for early therapeutic response, while the other OCT biomarkers, when taken separately, achieve a poor predictive value, almost equal to coin-tossing (Figure 5, Table 4).

Multiple logistic regression analysis showed that a predictive model for early therapeutic response after intravitreal anti-VEGF in terms of anatomical outcome included initial CMT, the presence of intraretinal cysts at baseline, and SII (Table 5).

The described model showed a very good predictive value (AUC ROC 0.911) with a sensitivity of 72.73% and a specificity of 100% at a cut-off value of >0.754 (Figure 6).

However, the defined model had a poor predictive value for functional response (AUC ROC 0.622; sensitivity 79.17%; specificity: 50%).

A separate regression analysis was conducted to investigate potential predictors for the change in BCVA after 4 weeks at follow-up in the study group. Interestingly, initial BCVA, initial CMT, age, and glycemic status were not related to the early functional response. Lower values of PLR (r:−0.27; *p* = 0.004) and the absence of HRS at baseline (r:−0.3; *p* = 0.001), but not of DRIL (r:0.17; *p* = 0.05) and SRF (r:0.22; *p* = 0.02), were correlated with early functional improvement (Table 6).

The model showed a good predictive value (AUC ROC: 0.818), with 79.2% sensitivity and 82.1% specificity (Figure 7).

## 4. Discussion

This study provides valuable insights into the potential predictors of response to anti-VEGF therapy in patients with DME. The results highlight that both baseline OCT biomarkers and blood cell-derived systemic inflammatory indices add valuable information regarding the early response to anti-VEGF therapy and should be included in a personalized therapeutic approach. Besides specific diabetic metabolic changes, abnormalities in the retinal microvasculature, oxidative stress, and low-grade systemic inflammation play a significant role [37,38,39,40]. Multiple cytokines and chemokines, such as IL-1β, IL-6, IL-8, TNF-α, and MCP-1, were found to be increased in serum and vitreous of the patients with diabetic retinopathy and DME [41,42]. However, implementing these findings in clinical practice is challenging, especially in low- to middle-income countries. Blood-based systemic inflammation biomarkers are easily available from a complete blood count and could offer valuable clinical information.

Several studies proved that NLR, PLR, and SII are higher in diabetic patients, compared to healthy controls [43,44,45]. Moreover, they correlate with the severity of diabetic microvascular complications, including diabetic retinopathy [46,47,48]. In a study involving 37 naive patients, Liao et al. [49] found that NLR and SII were significantly higher in patients with CMO vs. diffuse retinal thickness, with a “cut-off” value of 2.27. No correlations were found with changes in BCVA and CMT after anti-VEGF therapy. In a recent study by Zhou et al. [50], the number of retinal HRFs in diabetic macular edema correlated with the level of blood inflammatory markers. This finding was considered to support the theory of HRF’s inflammatory nature and the role of inflammation in DME [50].

However, to our knowledge, there is limited evidence regarding the predictive value of the blood cell-derived inflammation markers for the response to anti-VEGF therapy in patients with DME. The limited number of patients included and the different anti-VEGF agents used may explain the conflicting results obtained by various authors. Hu et al. [51], in a study involving 91 naïve patients with DME treated with intravitreal Ranibizumab, found that a pretreatment of NLR ≥ 2.27 was associated with inferior BCVA outcomes. Another recent research study by Katic et al. [52] involving 78 patients treated with bevacizumab over a 4-month follow-up period described significantly higher NLR, PLR, MLR, and SII in non-responders, with NLR and SII exhibiting the best predictive value (AUC ROC of 0.778 and 0.709, respectively). Moreover, they found that patients who responded to the therapy had worse baseline BCVA and higher values of CMT compared to non-responders [47]. Another study by Yalinbas and col. [53], comparing NLR and monocyte-to-high density lipoprotein in diabetic patients with and without DME, revealed that a “cut-off” value of NLR ≥ 2 is highly associated with DME, with an area under the ROC curve of 0.72. However, the authors did not find a correlation between pretreatment NLR and functional or anatomical outcomes [53].

In the present study, we found significantly higher pretreatment values for NLR, PLR MLR, and SII in non-responders, compared to the patients who experienced significant anatomical improvement after IVT aflibercept. Moreover, we found the best predictive value for NLR at a cut-off value of 2.32, (area under ROC curve of 0.778), similar to Hu [51] and Katic [52]. We also found that SII may be a useful predictive tool, with an area under the ROC curve of 0.788 for a cut-off value of 543.33.

Moreover, early therapeutic response correlates with baseline retinal morphology and specific OCT biomarkers. Baseline CMT emerged as a strong predictor of early anatomical improvement. In the present study, we found that higher initial CMT > 388 µm, SRF, and IRC were associated with favorable outcomes. This finding is consistent with previous studies that demonstrated a positive association between baseline macular edema severity and the magnitude of treatment response [12,22,23,52,54]. For instance, Schreur et al. [22] highlighted the predictive value of baseline CMT in determining the extent of anatomical recovery, suggesting that a higher initial burden may allow for more substantial reductions [22]. The significance of baseline CMT as a predictive factor aligns with findings by Vujosevic et al. [1], who highlighted the relationship between baseline anatomic characteristics and outcomes of anti-VEGF therapy in clinical practice. Similar to Gurung et al. [54], in the present study, a better BCVA before treatment was associated with a limited therapeutic response.

Previous studies reported controversial results regarding the value of HRF in early therapeutic response. While Huang [55] and Scheur [56] found positive correlations with early visual gain, other authors found that preexisting HRF is a negative predictive factor [57,58]. The morphological signification of HRF may differ, according to their localization in the inner or outer retinal layers. Several possible origins have been proposed, including subclinical hard exudates, activated glial cells, degenerated photoreceptors, over-phagocytosed RPE, or RPE metaplasia [55,57]. Moreover, a recent study by Zhou et al. [50] found a significant correlation between the number of HRF and the systemic inflammation markers NLR, PLR, and SII, supporting the role of inflammation in the etiology of HRF [50]. In our study, HRF presence was not correlated with anatomical improvement (r = 0.04; *p* = 0.68) but was a negative predictor for early vision improvement in a multivariate analysis (*p* = 0.04).

DRIL was identified as a significant biomarker associated with poorer visual outcomes, corroborating findings from previous investigations [11,23]. Sun et al. [23] highlighted that DRIL reflects underlying retinal damage and disruption of neural connectivity, which limits the potential for functional recovery despite anatomical improvements [1,59,60]. Moreover, other studies found that DRIL may reflect capillary non-perfusion in the macular region [61,62] and has been considered a non-invasive marker for macular ischemia. However, DRIL can resolve—either early or delayed—after anti-VEGF treatment, which is usually associated with a significant gain in BCVA [62,63]. In the present study, we found a low positive correlation (r = 0.25; *p* = 0.009) between preexisting DRIL and gain in BCVA. Our findings support this understanding by suggesting that pretreatment DRIL may correlate with functional improvement if resolved during the follow-up period. This could aid clinicians in setting realistic expectations and considering adjunctive treatment strategies.

To our knowledge, this is the first study that aims to integrate both systemic and OCT-based biomarkers in a multivariate analysis that aims to predict early response after anti-VEGF therapy in DME.A multivariate model, including SII, initial CMT, and the presence of IRC, proved to be an excellent predictor for early anatomical response (area under ROC curve of 0.911), indicating the role of chronic low-grade systemic inflammation in the pathology of diabetic microvascular complications, including diabetic retinopathy with DME. However, our study also highlights that anatomical and functional outcomes are not predicted by the same elements, suggesting the potential of other OCT biomarkers, such as HRF and DRIL, as predictors of functional response. A recent article by Rai and col. strengthens the concept of functional diabetic retinopathy (FDR) [64], pointing out the role of diabetic retinal neurodegeneration (DRN) as a cause of functional deficit in patients with DME. However, the diagnosis of DRN may be impaired by the presence of macular edema, which masks the neuroretina thinning. Multifocal functional tests, rather than BCVA, could offer a more adequate perspective of retinal visual function [64].

The heterogeneity in treatment response observed in this study further underscores the need for personalized treatment strategies. While all patients in this cohort received aflibercept, comparisons with other anti-VEGF agents in prior studies, such as ranibizumab and bevacizumab [51,52,53,54], reveal similarities in predictors of response, but variability in clinical outcomes. The higher binding affinity of aflibercept for VEGF-A and placental growth factor (PIGF) may confer advantages in specific subgroups, such as those with persistent edema despite prior treatments [65,66,67]. However, the predictors of response identified in this study suggest that baseline anatomic and functional characteristics, rather than the specific anti-VEGF agent used, may have a more substantial impact on outcomes. Future studies directly comparing predictors of response across different anti-VEGF agents could further refine treatment algorithms.

Moreover, the clinical implications of these findings are significant. Identifying patients likely to respond to anti-VEGF therapy can optimize resource allocation and improve patient outcomes, particularly in settings where treatment accessibility is limited. Including OCT biomarkers as routine evaluation tools could enhance the precision of clinical decision-making, enabling earlier identification of non-responders who may benefit from alternative therapies or combination approaches. Furthermore, the findings highlight the need for continued research into adjunctive therapies targeting inflammation and neuroprotection to complement the vascular benefits of anti-VEGF agents. Emerging therapies, including intravitreal corticosteroids and novel molecular inhibitors, hold promise in addressing the multifactorial pathophysiology of DME [6,66,67].

The present study has some limitations. While this study provides valuable insights, its retrospective design may be a source of potential bias. Additionally, data regarding the length of diabetes, associated complications, therapy, or lipid profile were not available in all cases. Another limitation is the relatively short follow-up period limit. However, multiple authors report that one month of evaluation after the first anti-VEGF intravitreal injection is a valuable indication of follow-up response at 3 months [15,16].

Future prospective studies with longer follow-ups and larger sample sizes could better elucidate the long-term effects of aflibercept on both structural and functional outcomes. As DME also affects extra-foveal retinal function, the use of multifocal functional testing in DR management would better align with the new concept of functional diabetic retinopathy. Additionally, incorporating advanced imaging modalities, such as OCT-angiography (OCTA), could provide further insights into the interplay between microvascular changes and treatment response [68,69,70]. The integration of OCTA, which provides detailed vascular imaging, may improve our understanding of the vascular changes underlying treatment response. This recommendation aligns with growing evidence supporting the role of advanced imaging in predicting and monitoring treatment outcomes [69,70]. Exploring the role of systemic factors, such as inflammatory markers and metabolic control, in conjunction with exploring the potential of combination therapies, may also enhance our understanding of DME pathophysiology and treatment optimization.

## 5. Conclusions

This study highlights the prognostic value of systemic inflammatory indices and OCT-derived biomarkers in predicting early response to anti-VEGF therapy in patients with DME. Lower values of NLR and SII, higher baseline CMT, and the presence of IRC were independent predictors for early anatomical improvement, while higher PLR and HRF were adverse factors for rapid vision gain. These findings underscore the critical role of baseline inflammatory and anatomical parameters in identifying patients less likely to respond to anti-VEGF treatment. By integrating these predictive factors into routine clinical practice, clinicians can better stratify patients and develop more personalized treatment strategies, ultimately improving therapeutic outcomes for individuals with DME.

## Figures and Tables

**Figure 1 biomedicines-13-01308-f001:**
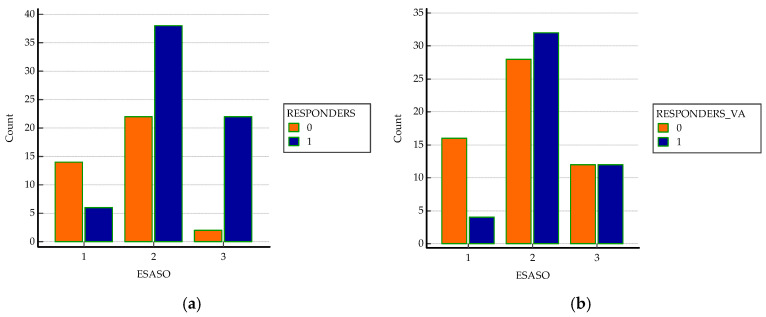
Correlations between ESASO grade and (**a**) early anatomical responders (CMT decrease ≥ 10%) and (**b**) early functional responders (BCVA change ≥ 0.1).

**Figure 2 biomedicines-13-01308-f002:**
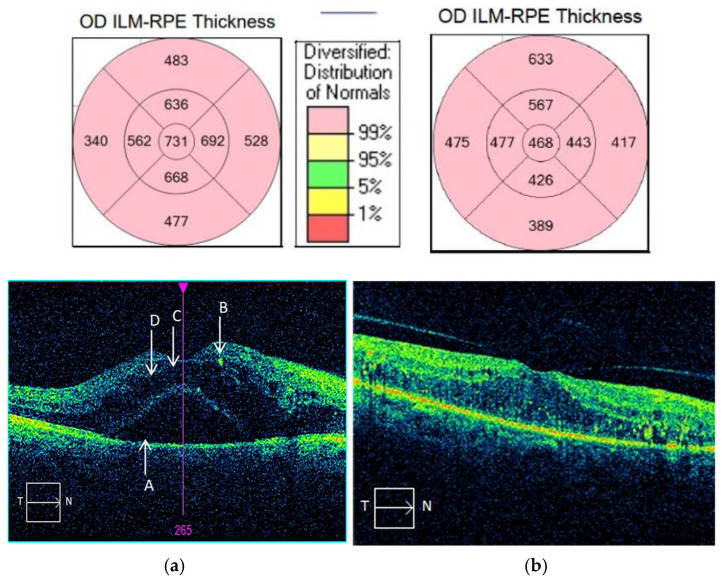
(**a**,**b**) OCT exam OD horizontal macular B-scan of an early “responder” patient before and one month after intravitreal aflibercept injection. A. Subretinal fluid (SRF). B. Hyperreflective retinal spots (HRSs). C. Intraretinal cysts (IRCs). D. Disorganization of retinal inner layers (DRILs); the comparative values are given in µm.

**Figure 3 biomedicines-13-01308-f003:**
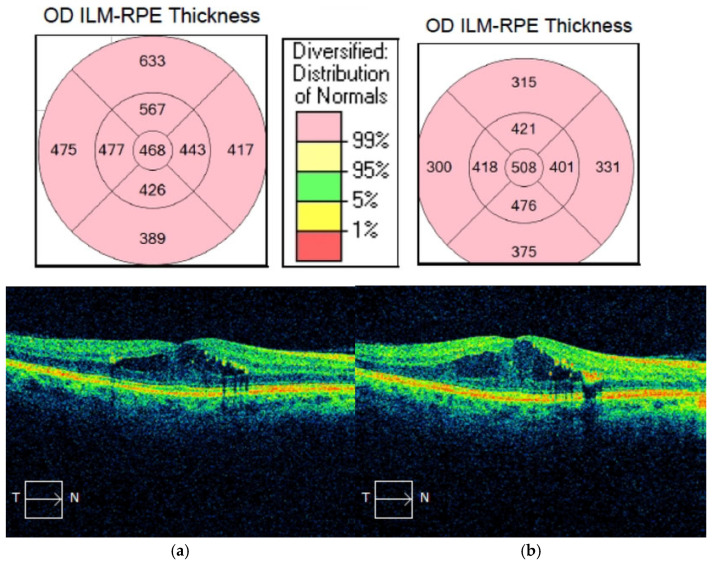
(**a**,**b**). OCT exam OD horizontal macular B-scan results of an “non-responder” patient before and one month after intravitreal aflibercept injection.; the comparative values are given in µm.

**Figure 4 biomedicines-13-01308-f004:**
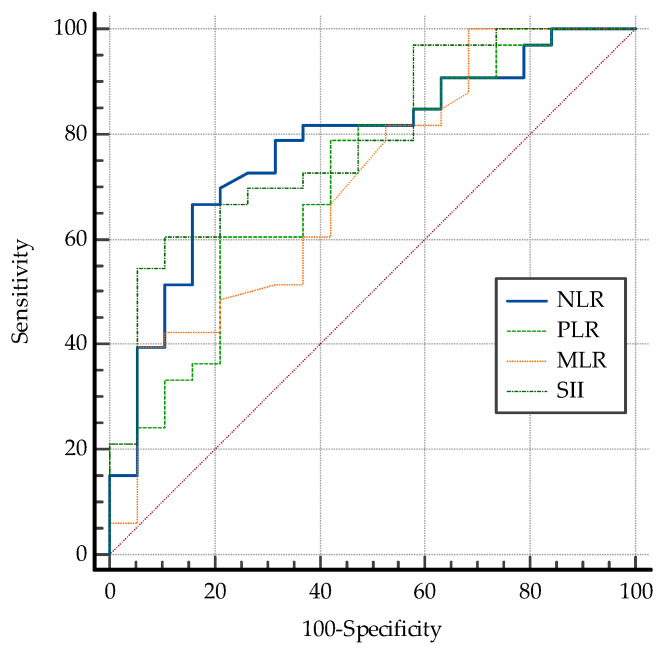
Comparative ROC curves for NLR, PLR, MLR, and SII for early response to anti-VEGF therapy in DME. NLR = neutrophil-to-lymphocyte ratio; MLR = monocyte-to-lymphocyte ratio; PLR= platelet-to-lymphocyte ratio; SII = systemic immune-inflammation index; DME: diabetic macular edema; red dashed line: the ROC curve for a random guess.

**Figure 5 biomedicines-13-01308-f005:**
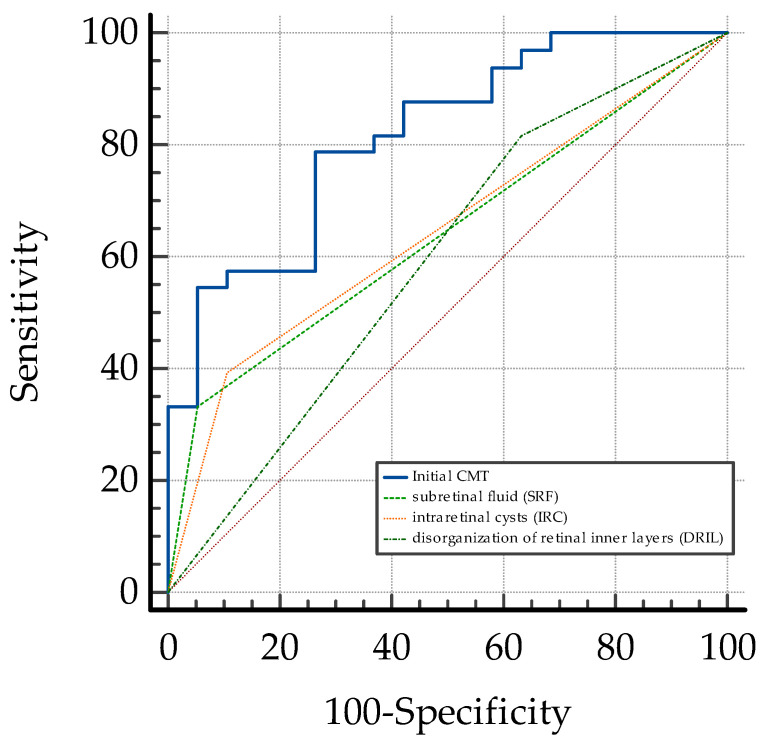
Comparison of ROC curves for OCT biomarkers: initial CMT, SRF, IRC, and DRIL.CMT= central macular thickness; SRF: subretinal fluid; IRCs: intraretinal cysts; DRIL: disorganization of retinal inner layers.; red dashed line: the ROC curve for a random guess.

**Figure 6 biomedicines-13-01308-f006:**
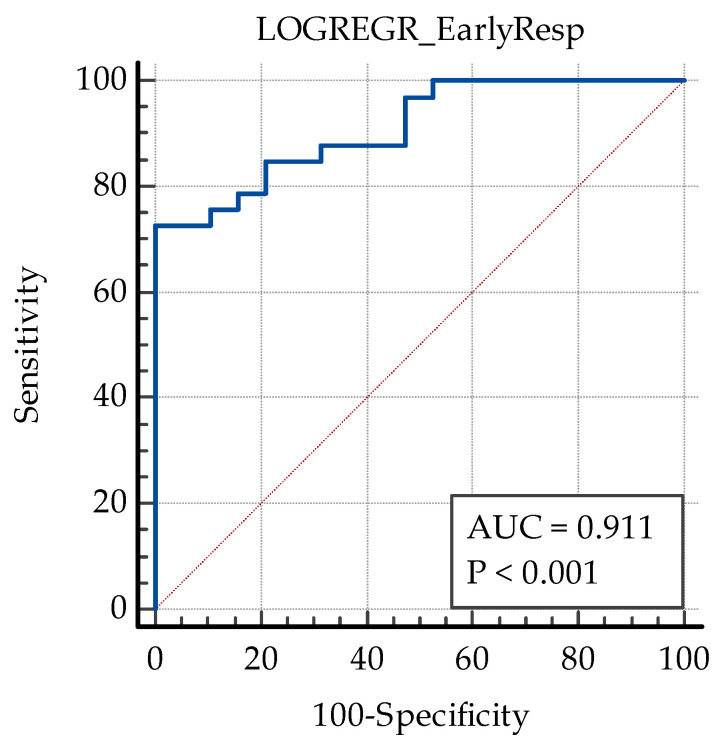
ROC curve of the predictive model for early anatomical response to anti-VEGF therapy. AUC: area under curve.; blue line: ROC curve for the regression model; red dashed line: the ROC curve for a random guess.

**Figure 7 biomedicines-13-01308-f007:**
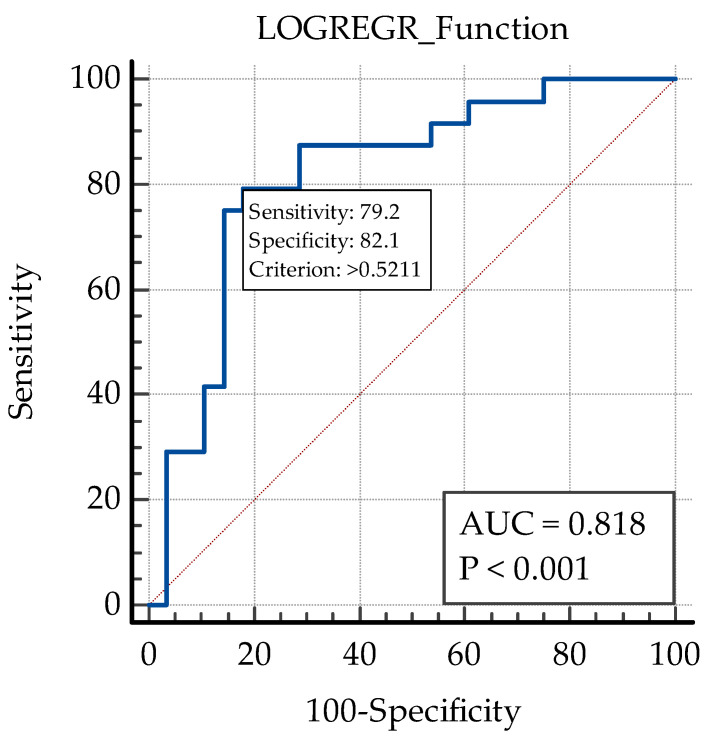
ROC curve analysis of the predictive model for early functional response after intravitreal anti-VEGF therapy in the study group. AUC: area under curve.

**Table 1 biomedicines-13-01308-t001:** Baseline characteristics of study population based on response to intravitreal anti-VEGF injections.

Variable	Early Responders * N = 66	Non-Responders ** N = 38	*p*-Value
Age (years, mean ± SD)	64.39 (±7.39)	63.00 (±8.35)	0.312 ^b^
Gender (N, %) M F	32(48.48%) 34 (51.52%)	26 (68.42%) 12 (31.58%)	0.077 ^a^
Eye (N, %) OD OS	30 (60.61%) 26 (39.39%)	18 (21.05%) 30 (78.95%)	0.081 ^a^*
DR Grading (N, %) NPDR PDR PPRD	54 (81.8%) 10 (15.15%) 2 (3.03%)	34 (89.47%) 4 (10.53%) 0 (0.0%)	0.529 ^c^
Neutrophils (cells × 10^9^/L, mean ± SD)	4.64 (±1.26)	5.47 (±1.44)	0.005 ^b^*
Monocytes (cells × 10^9^/L, mean ± SD)	0.58 (±0.17)	0.62 (±0.16)	0.116 ^b^
Lymphocytes (cells × 10^9^/L, mean ± SD)	2.19 (±0.73)	1.77 (±0.59)	0.005 ^b^*
Platelets (cells × 10^9^/L, mean ± SD)	246.27 (±55.09)	264.47 (±77.0)	0.443 ^b^
NLR (mean ± SD)	2.29 (±0.84)	3.4 (±1.44)	<0.001 ^b^*
MLR (mean ± SD)	0.28 (±0.09)	0.391 (±0.16)	<0.001 ^b^*
PLR (mean ± SD)	121.39 (±38.35)	164.92 (±71.46)	<0.001 ^b^*
SII (mean ± SD)	546.86 (±183.27)	886.44 (±407.06)	<0.001 ^b^*
Blood Glucose (mg/dL, mean ± SD)	154.36 (±55.58)	157.84 (±61.15)	0.668 ^b^
Urea (mg/dL, mean ± SD)	46.79 (±15.29)	48.84 (±20.58)	0.853 ^b^
Creatinine (mg/dL, mean ± SD)	0.98 (±0.33)	1.06 (±0.41)	0.385 ^b^

Footnote: * Early responders: ≥10% reduction in central macular thickness (CMT) from baseline at 30 days follow-up after the initiation of anti-VEGF therapy. ** Non-responders: <10% reduction in central macular thickness (CMT) from baseline at 30 days follow-up after the initiation of anti-VEGF therapy. ^a^ *p*-value was calculated by chi-squared test; ^b^ Mann–Whitney test; alpha risk was set to 5% (α = 0.05); ^c^ Fischer exact test; * *p* value <0.05 was considered statistically significant. Abbreviations: OD = right eye (oculus dextrus); OS = left eye (oculus sinister); DR: diabetic retinopathy; NPDR  =  non-proliferative diabetic retinopathy; PDR  =  proliferative diabetic retinopathy; PPDR = pre-proliferative diabetic retinopathy; NLR = neutrophil-to-lymphocyte ratio; MLR = monocyte-to-lymphocyte ratio; PLR = platelet-to-lymphocyte ratio; SII = systemic immune-inflammation index.

**Table 2 biomedicines-13-01308-t002:** Ocular characteristics at baseline and follow-up exam of study population based on response to intravitreal anti-VEGF injections.

Variable	Early Responders * N = 66	Non-Responders ** N = 38	*p*-Value
Initial BCVA (mean ± SD)	0.33 (±0.28)	0.50 (±0.33)	0.012 ^b^*
Post-in BCVA (mean ± SD)	0.48 (±0.33)	0.56 (±0.33)	0.193 ^b^
∆BCVA (mean ± SD)	0.15 (±0.17)	0.06 (±0.16)	0.003 ^b^*
Functional responders (N,%)	42 (63.6%)	10 (26.3%)	<0.001 ^a^*
Initial CMT (µm, mean ± SD)	507.42 (±148.52)	354.95 (±87.43)	<0.001 ^b^*
Post-inj CMT (µm, mean ± SD)	334.39 (±88.54)	351.16 (±94.73)	0.367 ^b^
∆CMT (µm, mean ± SD)	−173.03 (±127.65)	−3.79 (±15.58)	<0.001 ^b^*
∆CMT(%, mean ± SD)	−31.31 (±15.68)	−1.54 (±4.23)	<0.001 ^b^*
HRS (N, %)	32 (48.48%)	20 (52.63%)	0.839 ^a^
SRF (N, %)	22 (33.33%)	2 (5.26%)	0.001 ^c^*
DRIL (N, %)	54 (81.82%)	24 (63.16%)	0.06 ^a^
IRC (N, %)	26 (39.39%)	4 (10.53%)	0.004 ^a^*
ESASO grade (N, %) Early Advanced Severe Atrophic	6 (9.09%) 38 (57.6%) 22 (33.3%) 0 (0%)	14(36.8%) 22 (57.9%) 2 (5.3%) 0 (0%)	<0.0001 ^a^*

* Early responders: ≥10% reduction in central macular thickness (CMT) from baseline at 30 days follow-up after the initiation of anti-VEGF therapy. ** Non-responders: <10% reduction in central macular thickness (CMT) from baseline at 30 days follow-up after the initiation of anti-VEGF therapy. ^a^ *p*-value was calculated by chi-squared test; ^b^ Mann–Whitney test; alpha risk was set to 5% (α = 0.05); ^c^ Fischer exact test; * *p* value <0.05. This value was considered statistically significant for all tests. Abbreviations: Initial BCVA = baseline best-corrected visual acuity; post-inj BCVA = final best-corrected visual acuity; ∆BCVA = best-corrected visual acuity change between final and baseline; initial CMT = baseline central macular thickness; post-inj CMT = final central macular thickness; ∆CMT = central macular thickness change between final and baseline; ∆CMT(%) = ∆CMT/initial CMT × 100.; HRS: hyperreflective spots; SRF: subretinal fluid; DRIL: disorganization of retinal inner layers; IRC: intraretinal cysts; SD: standard deviation; ESASO grading: The European School for Advanced Studies in Ophthalmology Classification for DME.

**Table 3 biomedicines-13-01308-t003:** Comparative predictive value of NLR, MLR, PLR, and SII in study group.

Variable	AUC	SE	95%CI	Specificity	Sensitivity	Cut-Off Value
NLR	0.778	0.0471	0.685 to 0.853	84.2	66.67	≤2.32
PLR	0.719	0.0522	0.623 to 0.803	78.9	60.5	≤120.55
MLR	0.704	0.0535	0.607 to 0.790	94.7	39.4	≤0.21
SII	0.788	0.0451	0.697 to 0.862	89.5	60.6	≤543.53

Footnote: AUC: area under curve; SE: standard error; CI: confidence interval. NLR = neutrophil-to-lymphocyte ratio; MLR = monocyte-to-lymphocyte ratio; PLR= platelet-to-lymphocyte ratio; SII = systemic immune-inflammation index.

**Table 4 biomedicines-13-01308-t004:** Predictive value of CMT, SRF, IRC, and DRIL.

Variable	AUC	SE	95% CI	Specificity	Sensitivity	Cut-Off Value
Initial CMT	0.818	0.0415	0.730 to 0.887	73.70	78.80	>388
SRF	0.640	0.0345	0.540 to 0.732	94.71	33.30	>0
IRC	0.644	0.0394	0.544 to 0.736	89.47	39.39	>0
DRIL	0.593	0.0463	0.493 to 0.689	36.84	81.82	>0

Footnote: AUC: area under curve; SE: standard error; CI: confidence interval; HRSs: hyperreflective spots; SRF: subretinal fluid; DRIL: disorganization of retinal inner layers; IRC: intraretinal cysts; initial CMT: baseline central macular thickness.

**Table 5 biomedicines-13-01308-t005:** Logistic regression model for early responders to anti-VEGF therapy.

Variable	Coefficient	Std. Error	Wald	*p*
Initial CMT	0.011465	0.0033608	11.6368	0.0006
SII	−0.0063546	0.0016831	14.2546	0.0002
IRC	2.05223	0.95954	4.5743	0.0325
Constant	−0.32010	1.40766	0.05171	0.8201

Footnote: SII = systemic immune-inflammation index; IRCs: intraretinal cysts; initial CMT: baseline central macular thickness.

**Table 6 biomedicines-13-01308-t006:** Description of logistic regression model for early BCVA improvement in study group.

Variable	Coefficient	Std. Error	Wald	*p*
PLR	−0.014675	0.0051203	8.2142	0.0042
HRS	−1.77681	0.49220	13.0318	0.0003
DRIL	1.16247	0.55797	4.3405	0.0372
SRF	1.16709	0.55406	4.4370	0.0352
Constant	1.58203	0.92866	2.9021	0.0885

Footnote: PLR = platelet-to-lymphocyte ratio; HRSs: hyperreflective spots; SRF: subretinal fluid; DRIL: disorganization of retinal inner layers.

## Data Availability

Data is contained in the paper.

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
