# Peer review of "Predictive Role of Complete Blood Count-Derived Inflammation Indices and Optical Coherence Tomography Biomarkers for Early Response to Intravitreal Anti-VEGF in Diabetic Macular Edema"

_biomedicines, 2025, doi:10.3390/biomedicines13061308_

Round 1
Reviewer 1 Report
Comments and Suggestions for Authors
The authors aimed “to evaluate the predictive value of complete blood count (CBC)-based inflammation indexes and optical coherence tomography (OCT) parameters in determining early anti-VEGF treatment effectiveness in DME.”
Introduction
The introduction is adequate and supports the relevance of the study, highlighting the importance of correlating imaging biomarkers with data obtained through blood analysis. Interestingly, the authors could contextualize the possible inflammatory effect described in European patients showing transcriptional expression pattern differences that align with the anti-VEGF response in diabetic macular edema (https://pmc.ncbi.nlm.nih.gov/articles/PMC10590013/).
Methods
The methodology is clearly described; however, if the study was retrospective, why was only one post-treatment timepoint (1 month) included?
How many patients with type 1 diabetes mellitus (T1DM) were included, and was their inclusion justified?
Another ethical aspect concerns the blood collection described in this retrospective study. Why was blood drawn during the standard intravitreal anti-VEGF injection procedure?
Results
The authors initially state they studied early “responders” based on a ≥10% reduction in central macular thickness (CMT) from baseline. However, the tables refer to “responders” and “non-responders” without further clarification. The authors should consider, for instance, the definitions used in DRCR.net Protocol I (https://pubmed.ncbi.nlm.nih.gov/27644589/), which include CMT reduction and/or best corrected visual acuity (BCVA) improvements at least 180 days after treatment.
The p-value for CMT in Table 2 appears incorrect (reported as 0.502). Additionally, the value of 616 μm mentioned in line 214 is not found in the table.
If non-responders had lower baseline CMT, fewer cases of subretinal fluid (SRF), and intraretinal cysts (IRC), could these be cases with an atrophic phenotype? See: https://link.springer.com/article/10.1007/s00417-024-06473-2.
For scientific clarity, it would be useful for the authors to include an image example of a non-responder case.
These are more clinical aspects that were neither discussed nor detailed in a study claiming to assess anti-VEGF treatment effectiveness in DME. However, what type of DME was being evaluated?
Author Response
Dear Reviewer, many thanks for your time helpful comments that helped us improve our manuscript. We have carefully revised our paper and we have done the required corrections.
Reviewer 1:
The authors aimed “to evaluate the predictive value of complete blood count (CBC)-based inflammation indexes and optical coherence tomography (OCT) parameters in determining early anti-VEGF treatment effectiveness in DME.”
Introduction
The introduction is adequate and supports the relevance of the study, highlighting the importance of correlating imaging biomarkers with data obtained through blood analysis. Interestingly, the authors could contextualize the possible inflammatory effect described in European patients showing transcriptional expression pattern differences that align with the anti-VEGF response in diabetic macular edema (https://pmc.ncbi.nlm.nih.gov/articles/PMC10590013/).
R: Thank you for the suggestion. We have added this info in the introduction, and updated the references accordingly.
Methods
The methodology is clearly described; however, if the study was retrospective, why was only one post-treatment timepoint (1 month) included?
R: While routine protocols require at least 5 monthly anti-VEGF injections, with reevaluation of the response at 180 days, there is an increased interest for early prediction, taking into account that up to 40% of the patients do not reach an optimal response. This prediction could improve the patient-doctor relationship, because we can better discuss with the patient what we expect it will happen in the next weeks or months, during therapy. In the present study we intended to analyze the predictive factors for early therapeutic response, at 4 weeks after the first anti-VEGF injection. Please consider our results as a complimentary tool, not a definitive answer. In conjunction with other clinical assessments, it may help the provider to estimate the outcomes and prepare the patient for a potential delayed response, or the necessity to consider alternate treatment options, a worthy advantage in the effort to optimize visual preservation in DME patients.
How many patients with type 1 diabetes mellitus (T1DM) were included, and was their inclusion justified?
R: Thank you for the observation, we corrected. Only T2DM patients were included in the study.
Another ethical aspect concerns the blood collection described in this retrospective study. Why was blood drawn during the standard intravitreal anti-VEGF injection procedure?
R: The study received our hospital Ethical Committee approval. As our diabetic patients are not always fully investigated when they address to ophthalmologic evaluation, we perform a brief paraclinical exam at the treatment initiation, including complete blood count, blood sugar, HbA1C, urea and creatinine, and a lipid profile. In cases with poor glycemic control, we send them to diabetic reevaluation. This procedure is not performed at every injection, only at baseline.
Results
The authors initially state they studied early “responders” based on a ≥10% reduction in central macular thickness (CMT) from baseline. However, the tables refer to “responders” and “non-responders” without further clarification. The authors should consider, for instance, the definitions used in DRCR.net Protocol I (https://pubmed.ncbi.nlm.nih.gov/27644589/), which include CMT reduction and/or best corrected visual acuity (BCVA) improvements at least 180 days after treatment.
R: We have added the required clarifications in the tables.
Dear reviewer, we totally agree that the outcomes of the anti-VEGF therapy should be evaluated at least 180 days after initiation. In the present study we intended to analyze the predictive factors for early therapeutic response, at 4 weeks after the first anti-VEGF injection. Please consider our results as a complimentary tool, not a definitive answer. In conjunction with other clinical assessments, it may help the provider to estimate the outcomes and prepare the patient for a potential delayed response, or the necessity to consider alternate treatment options, a worthy advantage in the effort to optimize visual preservation in DME patients.
The p-value for CMT in Table 2 appears incorrect (reported as 0.502). Additionally, the value of 616 μm mentioned in line 214 is not found in the table.
R: Thank you for the comment, we have corrected.
If non-responders had lower baseline CMT, fewer cases of subretinal fluid (SRF), and intraretinal cysts (IRC), could these be cases with an atrophic phenotype? See: https://link.springer.com/article/10.1007/s00417-024-06473-2.
For scientific clarity, it would be useful for the authors to include an image example of a non-responder case.These are more clinical aspects that were neither discussed nor detailed in a study claiming to assess anti-VEGF treatment effectiveness in DME. However, what type of DME was being evaluated?
R: Thank you for the helpful comments. We totally agree that the use of ESASO classification could bring more info regarding the DME types encountered in the study group, rather the individual OCT biomarkers. We re-analyzed the data of the patients included in the study and classified the baseline OCT aspect according to the ESASO Classification of the DME: as early, advanced, severe and atrophic. We further statistically analyzed the correlations between baseline type of DME and the anatomical and functional outcomes observed at follow-up. There weren’t patients with atrophic DME in our study group. We found that patients in early stages, with minimal increase in CMT and minimal decrease in BCVA, experienced lower therapeutic effects at 4 weeks follow-up, compared to those with advanced DME. However, functional gain was less in severe stages according to ESASO classification, probably due to long term degenerative changes.
We also added images for non-responders, and updated the references as required.
We hope in this revised version, you will find it suitable for publication.
Reviewer 2 Report
Comments and Suggestions for Authors
The paper is well written in a fluent english.
The argument is quite interesting
Recent studies suggest that CBC-based inflammation indexes, such as the Systemic Immune-Inflammation Index (SII), may have predictive value for persistent diabetic macular edema (DME). The SII, in particular, has been proposed as a tool to identify individuals at greater risk of diabetes-related complications, including DME. This highlights the potential utility of CBC-derived inflammatory markers in predicting the persistence and progression of diabetic macular edema
.
The author propose to evaluate SII and OCT biomarkers to predict response to ANTi-VEGF intravitrel therapy in patients with DME
Altough the concept of this paper is absolutely not original , due to many paper published in the past years, this idea is very interesting.
Based to SII levels and DME ultrastructure might be possible to recognize patients not responders to intravitreal anti-vegf and begin therapy with intravitreal steroids and methabilic adjustment .
I detected no plagiarism..
Author Response
Dear Reviewer,
Many thanks for your kind words and appreciation of our work!
Reviewer 3 Report
Comments and Suggestions for Authors
Specific comments:
Abstract
L25: Correct double dash: --
L26: Please do not start a sentence in numerical. Write it in words.
Introduction
DME is patchy in nature, and quite commonly it is extra-foveal in locations. So, how logical do you think to take only the CMT as the OCT parameter?
L86: Provide full form for OCT.
Methods
L102: You have already defined DME in Introduction. So just use abbreviation (DME) here.
Q: How did you define DME?
L103: This full form of OCT should be mentioned in L86 in Introduction, and mention only abbreviation here.
L106: should it be BCVA?
L108: DR was already abbreviated above, so write DR here.
L111: Write only OCT.
L114: Write DRIL only as defined above already.
L115: Write CMT only as defined above already.
N.B.: Please not to provide full form in the first mention with abbreviations within a bracket. After that please consistently use only the abbreviations throughout the text, except in Figures and Tables, where you need to provide full form separately. I am not addressing such mistakes below this.
L132: Define IOP. You have mentioned intraocular pressure in L106, but did not provide the abbreviation.
L142: DR
LL156: Since you have mentioned “functional response” here, it may be a great idea to briefly mention about the new concept of DR/DMO management (functional diabetic retinopathy) in Introduction or in Discussion: https://doi.org/10.1016/j.survophthal.2024.11.010
Results
L178: Start writing in Words, not numerical.
Please avoid mistakes of abbreviations. I see it throughout the text. You have defined above with abbreviations, but repeatedly you are writing in full forms. Please correct them all.
L215: In the Methods section under Inclusion criteria L103: you have mentioned that the DME was confirmed by the presence of SRF and IRF on OCT images. This means all included patients should have SRF or IRF. But here in Results, you have mentioned that the SRF and IRF associated with early responders. I advise you to re-write this sentence or modify your inclusion criteria.
Please provide full forms for all the abbreviations used in Tables Table titles, Figures and figure legends.
Discussion
L394: angio-OCT can be properly written as OCT angiography (OCTA).
Q: In the evaluation of DR/DME, BCVA covering only fovea is the only functional test done. DR/DME also affect extra-foveal retinal function. So it is important to include multifocal functional testing in the management as highlighted with the new concept of functional diabetic retinopathy. It is worthwhile to include it in the future direction.
Author Response
Dear Reviewer, many thanks for your time helpful comments that helped us improve our manuscript. We have carefully revised our paper and we have done the required corrections.
Abstract
L25: Correct double dash: --
R: We corrected.
L26: Please do not start a sentence in numerical. Write it in words.
R: We corrected
Introduction
DME is patchy in nature, and quite commonly it is extra-foveal in locations. So, how logical do you think to take only the CMT as the OCT parameter?
R: Thank you for this observation. We totally agree, that CMT may not be the most accurate to quantify it, but it is the most reproducible, and less operator dependent. In our study, though presenting some degree of eccentricity, DME involved fovea in all cases. We added this info in the Materials and methods. We also added this comment, as a limitation of the study in the Discussions.
L86: Provide full form for OCT.
We corrected.
Methods
L102: You have already defined DME in Introduction. So just use abbreviation (DME) here.
We corrected.
Q: How did you define DME?
R: We have added more information about this. We defined DME included in our study, as clinically significant central-involved DME associated with vision loss, diagnosed clinically and confirmed by the presence of fovea thickening of ≧250 µm, by OCT macular scan. Furthermore, based on the reviewers’ comments, we employed ESASO classification of DME to better characterize the types of DME encountered in the study group.
L103: This full form of OCT should be mentioned in L86 in Introduction, and mention only abbreviation here.
We corrected
L106: should it be BCVA?
Yes, we corrected.
L108: DR was already abbreviated above, so write DR here.
We corrected
L111: Write only OCT.
We corrected.
L114: Write DRIL only as defined above already.
We corrected.
L115: Write CMT only as defined above already.
We corrected.
N.B.: Please not to provide full form in the first mention with abbreviations within a bracket. After that please consistently use only the abbreviations throughout the text, except in Figures and Tables, where you need to provide full form separately. I am not addressing such mistakes below this.
R: Thank you for your time and advices. We have carefully revised our manuscript accordingly.
L132: Define IOP. You have mentioned intraocular pressure in L106, but did not provide the abbreviation.
We corrected
L142: DR
We corrected
LL156: Since you have mentioned “functional response” here, it may be a great idea to briefly mention about the new concept of DR/DMO management (functional diabetic retinopathy) in Introduction or in Discussion: https://doi.org/10.1016/j.survophthal.2024.11.010
R: Thank you for the suggestion. We added the idea in the Discussion and updated the reference list according to your suggestion.
Results
L178: Start writing in Words, not numerical.
R: We corrected.
Please avoid mistakes of abbreviations. I see it throughout the text. You have defined above with abbreviations, but repeatedly you are writing in full forms. Please correct them all.
R: We have carefully revised our manuscript for errors of abbreviations.
L215: In the Methods section under Inclusion criteria L103: you have mentioned that the DME was confirmed by the presence of SRF and IRF on OCT images. This means all included patients should have SRF or IRF. But here in Results, you have mentioned that the SRF and IRF associated with early responders. I advise you to re-write this sentence or modify your inclusion criteria.
R: Thank you, we have corrected. The inclusions criteria were: patients with treatment-naïve clinically significant central-involved DME associated with vision loss, diagnosed clinically and confirmed by the presence of fovea thickening by OCT macular scan.
Please provide full forms for all the abbreviations used in Tables Table titles, Figures and figure legends.
Discussion
L394: angio-OCT can be properly written as OCT angiography (OCTA).
We corrected.
Q: In the evaluation of DR/DME, BCVA covering only fovea is the only functional test done. DR/DME also affect extra-foveal retinal function. So it is important to include multifocal functional testing in the management as highlighted with the new concept of functional diabetic retinopathy. It is worthwhile to include it in the future direction.
R: Thank you for the suggestion. We added the idea in the Discussion and updated the reference list according to your suggestion.